# Rectal Cancer: Exploring Predictive Biomarkers Through Molecular Pathways Involved in Carcinogenesis

**DOI:** 10.3390/biology13121007

**Published:** 2024-12-03

**Authors:** Sheila Martins, Pedro Veiga, José Guilherme Tralhão, Isabel Marques Carreira, Ilda Patrícia Ribeiro

**Affiliations:** 1Portuguese Oncology Institute of Coimbra, 3000-075 Coimbra, Portugal; sbastosmartins@gmail.com; 2Faculty of Medicine, University of Coimbra, 3000-548 Coimbra, Portugaljgtralhao@ulscoimbra.min-saude.pt (J.G.T.); iribeiro@uc.pt (I.P.R.); 3Surgery Department, Unidade Local de Saúde de Coimbra (ULS Coimbra), 3004-561 Coimbra, Portugal; 4Coimbra Institute for Clinical and Biomedical Research (iCBR) and Center of Investigation on Environment Genetics and Oncobiology (CIMAGO), Faculty of Medicine, University of Coimbra, 3000-548 Coimbra, Portugal; 5Center for Innovative Biomedicine and Biotechnology (CIBB) and Clinical Academic Center of Coimbra (CACC), University of Coimbra, 3000-548 Coimbra, Portugal

**Keywords:** rectal cancer, biomarker, neoadjuvant chemoradiotherapy, radiosensitivity

## Abstract

Locally advanced rectal cancers are treated with chemoradiotherapy before surgery to cause tumor downsizing and downstaging, enabling the surgery to excise the entire tumor and reduce the risk of local relapse. Approximately 20% of rectal cancers completely disappear after chemoradiotherapy, allowing patients to participate in a close surveillance program and potentially avoid surgery. Conversely, between 20% and 38% of patients may experience either a residual response or growth of the cancer during chemoradiotherapy, indicating no advantage in postponing surgery. If a method could be developed to differentiate these tumors early on, it would represent a significant breakthrough in the treatment strategy for rectal cancer, enabling a more tailored therapeutic approach. This review aimed to identify potential molecular biomarkers that may predict tumoral response to chemoradiotherapy in locally advanced rectal cancer. Some molecular biomarkers involved in rectal cancer genesis and progression show potential in differentiating the good from bad responders to chemoradiotherapy. The effectiveness of chemoradiotherapy for rectal cancer is assessed through the degree of tumor downstaging observed in MRI scans and during the histopathological examination of surgical resections. In the future, liquid biopsy may play a significant role in this evaluation.

## 1. Introduction

In 2022, colorectal cancer (CCR) had the second-highest incidence in Europe, preceded only by breast cancer. One-third of the colorectal cancers (181,796) were found in the rectum [1].

In recent decades, advances in radiologic staging, surgical techniques, histologic assessment, and adjuvant treatments have revolutionized the treatment approach in rectal adenocarcinoma. The concept of surgical resection with Total Mesorectum Excision (TME) [2], as well as the implementation of neoadjuvant chemoradiotherapy (nCRT) in locally advanced rectal cancer (LARC) treatment, has significantly contributed to reducing the risk of local recurrence. According to the current European Society for Medical Oncology (ESMO) guidelines, nCRT is recommended for LARC with advanced T stages (T3–T4), positive nodal disease in the mesorectum (N1–N2), or threatened or invaded mesorectal fascia [3].

The response to nCRT is variable and unpredictable. Approximately 20% of patients treated with nCRT achieve a pathological complete response (pCR), which is associated with better long-term outcomes compared to those without a complete response. In contrast, 20–38% of individuals experience a residual response or disease progression. About half of the patients exhibit a significant but incomplete response [4,5].

After nCRT, patients who exhibit no detectable tumor during a digital rectal examination, rectosigmoidoscopy, and magnetic resonance imaging (MRI) are deemed to have achieved a clinical complete response (cCR). These patients may be suitable candidates for enrollment in a Watch-and-Wait (W and W) strategy [5]. This organ preservation program, with a rigorous surveillance strategy to detect local regrowth, has become very appealing as it allows patients to be spared the morbidities inherent to surgery. In the USA, the National Comprehensive Cancer Network (NCCN) guidelines included W and W as a therapeutic option for patients with cCR [6]. In Europe, such patients may be considered for W and W inclusion in prospective registries or clinical trials.

According to the International Watch-and-Wait Database Consortium results, 25% of patients selected for this non-surgical approach presented local regrowth, mainly during the first two years of follow-up [7]. Within the first three years of follow-up, predominantly 10% developed distant metastases. Still, this number rises to 24.1% among patients with local regrowth, an independent factor associated with worse distant metastases-free survival in the multivariable model [8].

While nCRT improved local control rates, no significant impact on disease-free survival (DFS) was seen. Longer intervals following nCRT may improve pCR rates, with unknown prognostic implications. However, delaying surgery and postoperative systemic adjuvant chemotherapy increases, respectively, surgical morbidity and the risk of subsequent distant metastases, which is the leading cause of death in LARC [9,10].

Total Neoadjuvant Therapy (TNT) has emerged with the primary objective of addressing micrometastases at an earlier stage. This innovative treatment approach involves the administration of neoadjuvant chemotherapy and radiotherapy and is currently being evaluated in several ongoing trials using different regimens. The secondary goals of TNT include enhancing compliance with systemic chemotherapy and facilitating clinical downstaging to enable organ preservation [10]. However, the recent publication of the 5-year follow-up results of the RAPIDO trial has raised additional concerns. The study revealed an increase in local recurrence rate in the TNT group (10% vs. 6%) related to higher rates of breached TME (21% vs. 4%) [11]. Radiotherapy may induce tumor fragmentation, in which small groups of viable tumor cells may persist up to 3 cm in all planes around the central ulcer. These residual cell clusters are not detectable on restaging imaging and are more likely to occur in tumors that were initially more advanced and larger. In such cases, surgical treatment might not be radical, resulting in local recurrence and distant metastases [12,13]. Tumors that fail to respond to neoadjuvant treatment may exhibit progression during the extended preoperative phase. Furthermore, patients with radioresistant tumors are subjected to unnecessary toxicity without deriving any clinical benefit [14]. It is advisable to undertake an early image assessment for response evaluation to identify such patient cohorts and subsequently modify and customize the treatment approach [11].

The identification of predictors of pathological response to nCRT in LARC would represent a crucial advancement in the management of these patients. It would facilitate the development of personalized therapeutic strategies, enhance the potential for achieving complete clinical responses, decrease local and distant recurrences, improve survival rates, and diminish the morbidity and costs associated with ineffective treatments.

Numerous investigations have been conducted to identify these predictors of pathological response, which may include clinicopathological, radiological, or molecular biomarkers derived from peripheral blood or tumor tissue [5].

Increasing knowledge about molecular pathways in rectal cancer (RC), the biological factors of radiation response, and their underlying molecular basis (DNA-damage response and repair mechanisms, intracellular signaling pathways, and the tumor microenvironment) has allowed the identification of potential biomarkers [15].

This study focuses on the molecular pathways associated with rectal cancer and aims to identify and examine the rationale for potential molecular biomarkers in LARC.

## 2. Molecular Pathways in Rectal Cancer

The molecular landscape of RC includes several distinct and overlapping pathways that contribute to its development, progression, and treatment resistance. Although RC is often grouped with colon cancer (CC), it has distinct molecular and biological characteristics despite sharing some molecular pathways involved in carcinogenesis (Figure 1) [16].

### 2.1. Chromosomal Instability (CIN)

Chromosomal instability is observed in approximately 70% of sporadic CRC, particularly in the distal colon, and is recognized as a hallmark of cancer that drives its development and progression [17]. CIN includes both whole chromosomal alterations and structural changes, which may result from errors in DNA replication and chromosome segregation during cell division [18,19]. These chromosomal abnormalities often include frequent allelic losses, leading to the inactivation of tumor suppressor genes, and copy number gains, which can result in the activation of oncogenes [18].

CIN analysis using array comparative genomic hybridization (aCGH) is a valuable tool for the characterization of the tumor’s molecular profile, contributing to the detection of recurrent chromosomal alterations that can influence patient diagnosis and prognosis [18].

Alterations involving key genes such as *APC*, *TP53*, and *KRAS* are frequently observed in rectal adenocarcinoma [17]. However, when compared to CRC, *TP53* mutations are more frequent in patients diagnosed with RC [17]. Genomic data from The Cancer Genome Atlas (TCGA), PanCancer Atlas, and MSK, Nature Medicine 2022 further corroborates these findings (Figure 2).

The genetic landscape of rectal cancer is marked by multiple deletions and duplications across various chromosomes, affecting both tumor suppressor genes and oncogenes. (Figure 3).

### 2.2. Microsatellite Instability (MSI)

Microsatellite instability is associated with the loss of the DNA mismatch repair system (MMR) caused by genetic and epigenetic alterations in MMR genes, leading to a hypermutable phenotype [20]. In MSI, key genes responsible for DNA repair—such as *MLH1*, *MSH2*, *MSH6*, and *PMS2*—are either mutated or silenced. Approximately 15% of all CRC exhibit MSI, with 3% being due to Lynch syndrome and 12% occurring sporadically [20,21,22].

The MSI role in disease progression and prognosis is complex, since in early-stage CRC it is associated with a more favorable prognosis and in metastatic cancer it is associated with a poor outcome [21]. In RC, MSI occurrence is less common [21]. Fernebro et al. (2002) reported that 10% of rectal cancers from young patients were MSI-high; however, all of these tumors showed loss of expression for MSH2, which suggests an underlying hereditary nonpolyposis colorectal cancer-causing mutation [23]. Swets et al. (2022) detected MSI in only 7% of cases [21]. In a study conducted by Hasan et al. (2020), 13% were MSI positive [24].

These results show that even though MSI is rare in RC, there are several cases that exhibit this characteristic. Therefore, it is essential to analyze microsatellite stability to understand how it affects patient prognosis and to better characterize this small subgroup of patients.

### 2.3. CpG Hypermethylation

Hypermethylation of CpG islands in promoter regions can lead to transcriptional silencing of genes that are crucial for maintaining normal cellular functions, including those involved in DNA repair, apoptosis, and cell cycle regulation. This silencing contributes to the accumulation of genetic mutations and the progression of cancer [25].

The CpG island methylator phenotype (CIMP) status is important for the diagnosis and prognosis of patients with CRC since CIMP-high colorectal tumors have a distinct clinical, pathological, and molecular profile [18]. Although CIMP tumors are predominantly found in the right colon, they can also appear in the rectum [18,25,26,27]. However, the clinical significance of CIMP tumors at this site remains poorly understood [26].

In a study by Laskar et al. (2014), CIMP-high and CIMP-low phenotypes were identified in 43.7% and 38.75% of RC cases, respectively, while 17.5% were classified as CIMP-negative [28]. In contrast, Exner et al. (2015) reported only 3.8% of cases as CIMP positive using a more stringent marker panel. However, when the classic CIMP maker was applied, it revealed a positivity of 19.23%, highlighting the importance of using the most suitable gene panel for accurate CIMP analysis [29].

These findings confirm that CIMP can indeed be present in RC. Thus, analyzing methylation patterns could significantly enhance the diagnosis, prognosis, and treatment strategies for patients with RC.

### 2.4. p53 Pathway

The p53 pathway is crucial for maintaining genomic stability and is often referred to as the “guardian of the genome”. This transcription factor, which functions as a tumor suppressor, plays a critical role in responding to cellular stress and DNA damage through its interactions with various proteins involved in cell cycle regulation and apoptosis [30,31].

The p53 activity must be tightly regulated to ensure cellular homeostasis, primarily through a negative feedback loop that maintains low levels of p53 in normal cells, mediated by its negative regulator MDM2. The role of p53 in cell cycle arrest is also crucial since it can induce the expression of certain genes, such as p21, that inhibits the cell cycle by blocking cyclin-dependent kinases (CKDs). Ultimately, if cellular damage is too severe to be repaired, p53 triggers programmed cell death by inducing the expression of pro-apoptotic genes [31].

During the progression of CRC from benign adenomas to malignant carcinomas, p53 mutations are commonly observed, occurring in approximately 60% of cases. These mutations can result in either loss or gain of function and are associated with tumor development and growth [32]. In RC, the frequency of alterations in the p53 pathway remains less well-characterized. However, Klump et al. (2004) identified a significant association between p53 expression and tumor location, finding that left-sided tumors were more likely to exhibit p53 overexpression [33]. A literature review by Iacopetta (2003) reported higher frequencies of p53 expression in the distal colon and rectum [34]. Furthermore, Petrisor et al. (2008) demonstrated that 53% of patients with rectal adenocarcinoma tested positive for p53 [35]. Mondaca and Yaeger (2019) also reported that *TP53* mutations were detected more frequently in the rectum compared to the proximal colon, with prevalence rates of 81% and 65%, respectively [17].

### 2.5. MAPK Signaling Pathway and BRAF Mutation

The MAPK signaling pathway is involved in several cellular processes such as apoptosis, cell proliferation, differentiation, and development.

This pathway involves various isoforms of RAS (H-RAS, N-RAS, K-RAS) and RAF, each playing unique roles and having oncogenic potential. RAS activation occurs when it binds to guanosine triphosphate (GTP), while its inhibition happens when it binds to guanosine diphosphate (GDP). Once activated, RAS triggers RAF kinase, which in turn activates the downstream MAPK signaling pathway [36]. The BRAF protein, a serine/threonine kinase belonging to the RAF family, is also implicated in CRC through its involvement in the MAPK pathway.

The *BRAF*V600E mutation is particularly relevant, as it activates the RAS/RAF/MAPK pathway, promoting cell proliferation and survival while inhibiting apoptosis [37]. Tamas et al. (2015) demonstrated that the frequency of *BRAF* mutations was higher in the proximal colon than in the distal colon and rectum [38]. Mondaca and Yaeger (2019) also concluded that the *BRAF*V600E is rare in RC and seen in less than 1% of cases [17]. Laskar et al. (2015) reported that *BRAF*V600E was detected in 2.5% of cases of the RC patients [28].

Additionally, Nfonsam et al. (2016) demonstrated that the MAPK signaling pathway is the most deregulated pathway in early-onset rectal tumors, whereas the PI3K/AKT signaling pathway is the most deregulated in late-onset rectal tumors [39]. Chatila et al. (2022) also concluded that the WNT, TP53, and RAS were the most frequently altered signaling pathways [40].

In contrast to *BRAF* mutations, *KRAS* alterations are frequently detected in RC. According to data from The Human Protein Atlas (https://www.proteinatlas.org/, accessed on 23 October 2024), KRAS staining was high in normal tissue but was either not detected or exhibited medium staining in rectal tumor tissue (Figure 4). These observations suggest that *KRAS* expression may be reduced or absent in tumor tissue. A study conducted by Jo et al. (2016) found that 43% of patients had *KRAS* mutations, with the majority occurring in codon 35 [41]. This finding is consistent with Chatila et al. (2022), who reported that 42% of patients had *KRAS* mutations [40].

### 2.6. PI3K Signaling Pathway

The PI3K/AKT pathway is a critical intracellular signaling pathway that regulates various cellular functions such as growth, proliferation, metabolism, and survival.

In RC, as well as other types of cancers, dysregulation of this pathway is commonly associated with tumor progression, resistance to therapy, and poor prognosis [42].

The PI3K/AKT/mTOR pathway is initiated by activation of receptor tyrosine kinases (RTKs) or other cell surface receptors, which activate PI3K (phosphatidylinositol 3-kinase). PI3K then phosphorylates PIP2 (phosphatidylinositol-4,5-bisphosphate) to generate PIP3 (phosphatidylinositol-3,4,5-trisphosphate), which facilitates the recruitment of protein kinase B (AKT). The tumor suppressor phosphatase and tensin homolog (PTEN) dephosphorylates PIP3 back to PIP2, thereby limiting PI3K/AKT signaling. Increased levels of PIP3 due to PI3K activation enhance AKT activation, driving downstream effects such as cell survival and growth. Key downstream targets of activated AKT include mammalian targets of rapamycin (mTOR) and FOXO [42,43].

According to data from The Human Protein Atlas, analysis using the same antibody for evaluating protein expression revealed that PTEN exhibited low staining in RC tissue, indicating higher expression in normal tissue (Figure 5). In contrast, PIK3CA showed medium to high staining in both normal and tumor tissues (Figure 5). FOXO1 staining was medium in normal tissue but not detected to medium in rectal adenocarcinoma. These findings suggest potential loss of *PTEN* and *FOXO1* expression in tumor tissue, which may contribute to disease development and progression. This is consistent with literature reports of *PTEN* gene alterations in rectal adenocarcinoma [44,45].

Data from TCGA, available on the UALCAN (https://ualcan.path.uab.edu/, accessed on 25 October 2024) platform, also indicate that *PTEN* and *FOXO1* are highly expressed in normal tissue compared to tumor tissue (Figure 6—[46,47]).

According to the MSK cohort data reported in Nature Medicine (2022), available on cBioPortal for Cancer Genomics [48,49,50], which included 692 rectal cancer samples, 4% exhibited *PTEN* alterations, primarily consisting of missense mutations, deletions, and truncating mutations. For *PIK3CA*, 16% of patients had missense or in-frame mutations [40].

### 2.7. Wnt/β-Catenin Pathway

Wnt signaling, mediated by a family of glycolipoproteins, plays a crucial role in regulating cell proliferation, polarity, and fate determination, both during embryonic development and in the maintenance of tissue homeostasis. This pathway involves a complex network of protein interactions (Figure 7) [51,52].

In the absence of Wnt ligand, cytoplasmic β-catenin forms a complex with proteins including Axin, APC, GSK3, and CK1, where it is sequentially phosphorylated by CK1 and GSK3. This phosphorylation targets β-catenin for ubiquitination by the E3 ligase β-Trcp, leading to degradation by the proteosome. As a result, β-catenin levels remain low in the cytoplasm, and Wnt target genes are kept repressed by the transcriptional repressor complex TCF-TLE/Groucho and histone deacetylases (HDAC). When Wnt ligands bind to receptors (FZD and LRP5/6), the phosphorylation and degradation of β-catenin are inhibited. β-catenin then accumulates in the nucleus, where it partners with TCF to activate Wnt-responsive genes [51]. The regulation of β-catenin through phosphorylation and proteasomal degradation is a critical mechanism of the Wnt pathway. Dysregulation of this pathway, particularly excessive accumulation of nuclear β-catenin, can lead to aberrant activation of Wnt target genes, contributing to the development and progression of diseases such as CRC [53].

Zhang et al. (2020) identified *APC*R876* as a significant mutation hotspot in RC when compared to distal colon cancer [54]. Similarly, Bai et al. (2015) reported mutations in the *CTNNB1* gene in RC, though at a lower frequency [55]. Notably, they also observed mutations in the *APC* gene in 16.5% of RC cases. Data from TCGA, available on the UALCAN platform, also indicates that *APC* gene expression is lower in RC when compared to normal samples (Figure 8—[46,47]).

### 2.8. TGF-β Signaling Pathway

The Transforming Growth Factor-β (TGF-β) signaling pathway is essential for regulating various cellular processes, including proliferation, differentiation, apoptosis, angiogenesis, and inflammation [56,57]. The TGF-β signaling pathway is initiated when TGF-β ligands bind to a heterotetrameric receptor complex. This complex phosphorylates a group of SMAD proteins, which serve as mediators of the pathway and are classified into three categories: receptor-regulated SMADs (R-SMADs), inhibitory SMADs (I-SMADs), and common-mediator SMADs (Co-SMADs). Phosphorylated R-SMADs form a complex with SMAD4, a Co-SMAD, and translocate to the nucleus, where they interact with various transcription factors to regulate the expression of target genes involved in cell cycle regulation, differentiation, and immune responses [57] (Figure 9). The TGF-β pathway is controlled by both positive and negative feedback mechanisms. I-SMADs, such as SMAD7, inhibit R-SMAD activation through binding to the TGF-β type I receptor, blocking downstream signaling [58].

Mutations in TGF-β receptors and SMAD proteins are common in CRC. However, according to Zhang et al. (2020), SAMD4 mutations were more frequent in RC than distal colon cancer [54].

## 3. Predictive and Prognostic Biomarkers

With the constant expansion of omics technologies, it is possible to detect novel biomarkers that can be applied in clinical practice and research, aiming to provide more personalized treatments for patients based on tumor molecular landscape [59]. In addition to improving therapies, biomarkers also contribute to the diagnosis and stratification of patients, enabling early detection of disease [60].

The molecular profile of RC is characterized by multiple alterations in various genes, including both tumor suppressors and oncogenes, with certain alterations potentially impacting patient survival and treatment response. Therefore, in this chapter, we will discuss potential biomarkers associated with RC and their impact on patient prognosis.

Some of these biomarkers include chromosomal instability (CIN), microsatellite instability (MSI), CpG island methylator phenotype (CIMP), and mutations in specific genes such as *TP53*, *KRAS*, *BRAF*, *SMAD4*, and *PTEN* (Table 1).

The influence of CIN on treatment response in RC is not extensively documented. However, emerging evidence suggests that CIN may serve as both a prognostic and predictive biomarker in this context [61,62].

The influence of MSI in RC remains unclear. While MSI implications are well-studied in proximal CRC, these findings cannot be directly applied to RC. MSI incidence is lower in RC, and the literature lacks consensus on its implications [21]. Charara et al. (2004) reported improved treatment responses in RC patients undergoing nCRT, whereas O’Connell et al. (2020) found no significant differences in complete response rates after nCRT [63,64]. More recently, Lee et al. (2022) concluded that MSI was not associated with response to nCRT in RC patients [65]. Even though these results seem unfavorable, it is important to consider that the number of patients included in these studies is limited, largely due to the low prevalence of MSI in RC. Therefore, multicenter studies are needed to fully understand and predict MSI influence in RC.

Regarding CIMP, the literature presents conflicting data, with studies reporting contradictory findings [26,27,29].

In contrast, *TP53* mutations are widely reported to be associated with poor prognosis, with most studies supporting its role as a predictive biomarker in RC [67,68,69,70,71].

*KRAS* and *BRAF* are two key oncogenes in the MAPK pathway, and numerous studies have reported an association between mutations in these genes and poor prognosis [71,73,74,75,76,77,78]. However, other studies have not found this association for *KRAS* mutations [72].

Mutations in the *SMAD4* gene are associated with a worse prognosis in patients with LARC and could serve as a predictive biomarker for resistance to nCRT [78].

The *PTEN* gene, which is involved in the PI3K/AKT/mTOR signaling pathway, is also associated with clinical outcomes in LARC patients undergoing nCRT followed by radical surgery. Genetic alterations within this pathway, including those in PTEN, may influence treatment response and prognosis [75,79]

The integration of this genetic data with transcriptomic and epigenetic analyses in rectal cancer research provides a comprehensive view of the molecular mechanisms underlying tumor development, progression, and treatment response, ultimately leading to the characterization of robust biomarkers [80,81]. In a recent pilot study, Cicalini et al. (2024) demonstrated the potential of multi-omics analysis to predict CRT response in patients with LARC [81].

Boldrini et al. (2024) combine gut microbiota and ctDNA information to explore the predictive performance. Although this study is still in its early stages and no results have been published to date, it presents a promising and groundbreaking approach. By integrating the various molecular levels and microbiome data as potential biomarkers, this study’s results can improve patients’ outcomes [82].

Jiang et al. (2022) further demonstrated the potential of multi-omics approaches to identify predictive biomarkers, utilizing genomic DNA and cell-free DNA (cfDNA). Their findings revealed that increased expression of genes such as *EGFR* and *HSP90AA1* was associated with higher sensitivity to chemotherapy, while copy number losses were predominantly observed in patients with poorer prognoses. Importantly, *EGFR* and *HSP90AA1* exhibited significant differences across multiple molecular layers, demonstrating their potential as key biomarkers for prognosis and treatment response [83]. 

The development of multi-omics approaches offers a unique opportunity to unravel the complex landscape of rectal cancer, clarifying the complex interactions of genetic, transcriptomic, metabolomic, and epigenetic factors. By integrating diverse layers of molecular data, these approaches not only enhance our understanding of predictive biomarkers but also pave the way for the development of more precise diagnostic tools and personalized therapeutic strategies.

## 4. Liquid Biopsy

Liquid biopsies have emerged as a promising tool in oncology, offering a non-invasive approach to obtain information about tumor biology by analyzing biomarkers present in body fluids, such as blood, cerebrospinal fluid, and urine. Unlike traditional tissue biopsies, which require invasive procedures and provide only limited information about the tumor due to intratumoral heterogeneity, liquid biopsies allow continuous monitoring of cancer progression. They also offer the potential to detect relapse after treatment and facilitate molecular characterization of tumors [43,84,85].

In RC, accurate diagnosis and effective monitoring of the disease are crucial to improving patient outcomes. Standard diagnostic techniques, including imaging and tissue biopsy, remain indispensable but often fail to show the molecular complexity of the tumor. As RC progresses and undergoes treatment, tumor cells, along with genetic material and other cellular components, are shed into the bloodstream. Circulating tumor cells (CTCs), cell-free nucleic acids, and extracellular vesicles present an opportunity to monitor the disease progression, improving patient stratification and possible treatment outcomes [85]. Most of the analysis relies on Next-Generation Sequencing (NGS) and digital droplet polymerase chain reaction (ddPCR) due to their high sensitivity [86]. NGS allows the detection of a wide range of genomic alterations. However, its use in clinical practice is challenging due to the long turnaround time and associated costs. When compared to NGS, PCR-based technologies only enable the targeted analysis, with a low cost and short turnaround time [87].

Patients with RC who undergo radical surgery face a considerable risk of distant recurrence within the first five years, largely due to the presence of micrometastatic disease. Therefore, a liquid biopsy might provide information on treatment response and prognosis, helping to identify patients with a high risk of recurrence [88].

In the context of RC, liquid biopsy is also useful for detection of minimal residual disease (MRD), as it can affect the outcomes of patients [88,89].

### 4.1. Circulating Tumor Cells

Circulating tumor cells (CTCs) were first documented by T.R. Ashworth in 1869 [90] and represent a population of highly dynamic cancer cells present in the peripheral blood with origin in the tumor.

The quantity and genomic content of these cells have been shown to correlate with the progression of the primary tumor, as they are associated with tumor invasion and metastasis [91,92]. However, CTCs in the bloodstream are quite rare, and it poses a challenge in their isolation. As a result, an initial enrichment step is often necessary. Isolation methods frequently rely on the expression of epithelial cell adhesion molecule (EpCAM), utilizing antibodies that specifically target surface antigens on CTCs. Some of the techniques include the conjugation of antibodies with magnetic nanoparticles and the use of microfluidics [43,91,92].

The value of CTC detection in other types of cancer is widely described, such as breast cancer, prostate cancer, colorectal cancer, and liver cancer [93,94,95,96]. Regarding the use of CTCs in clinical practice, the Food and Drug Administration has approved The CellSearch™ by Veridex. This method has been used in clinical practice in patients with metastatic breast, prostate, and colorectal cancer [97].

In RC, several studies have explored the role of CTCs in disease progression and treatment response [98,99,100,101,102].

Sun et al. (2013) detected CTCs in RC patients and found that CTC levels were significantly higher in patients with metastatic disease compared to those with local recurrence or stage II-III disease [98]. Similarly, Liu et al. (2023) detected CTCs in 91.6% of patients and reported that the pCR and cCR were more common in patients who experienced a decrease in CTC count to less than one cell after radiotherapy [99]. Bahnassy et al. (2020) identified a significant association between elevated CTC levels in LARC patients and factors such as lymph node ratio, distant metastasis, and increased mortality [100].

Silva et al. (2021) investigated CTC kinetics before and after nCRT, concluding that higher post-treatment CTC kinetics was associated with poorer DFS and OS [102]. Flores et al. (2019) observed that patients who responded to nCRT lacked the expression of two specific proteins on their CTCs at the start of follow-up, and they also found that CTC kinetics correlated with disease outcomes in LARC patients [101].

These studies demonstrate the potential of CTCs as prognostic biomarkers in RC. However, larger-scale studies are needed to draw more robust conclusions and to validate the application of CTC analysis in routine clinical practice.

### 4.2. Cell-Free Nucleic Acids

In addition to CTCs, circulating nucleic acids (cfDNA/cfRNA) and circulating tumor DNA (ctDNA) can also be isolated from blood samples. ctDNA, a component of the cell-free DNA (cfDNA), originates from tumor cells and enters the bloodstream as a result of cellular processes such as cell death and apoptosis. Given the relatively low concentration of ctDNA in circulation, sensitive techniques are required to detect genetic variants and copy number variations (CNVs) effectively. The two primary methodologies used for the evaluation of ctDNA are ddPCR and NGS [43,103]. ctDNA levels vary among patients depending on the disease progression, tumor heterogeneity, effects of the treatment, and individual patient characteristics. This variation of ctDNA concentration has been evaluated as a prognostic tool [103].

Since ctDNA constitutes only a small fraction of cfDNA, highly sensitive techniques are required to identify tumor-specific alterations. Mutations in ctDNA can be detected by PCR or NGS. However, PCR-based methodologies only detect mutations in specific genes, limiting their utility. On the other hand, NGS approaches offer high coverage and can be applied in a genome-wide manner [104,105]. In addition to these methods, methylation assays are highly applicable to detect methylation patterns in specific regions. The use of MRE-seq appears to be a promising methodology. Kwon et al. (2023) demonstrated the applicability of this approach, showing high accuracy in diagnosing and detecting global hypomethylation patterns in lung and colorectal cancer [105].

In rectal cancer, Shalaby et al. (2017) demonstrated the ability of *MGMT* and *ERCC1* methylation status to distinguish between benign and malignant rectal tumors [106].

Regarding the analysis of ctDNA as a biomarker in RC, Wang et al. (2021) conducted a prospective cohort study in which 119 patients with LARC were recruited and ctDNA monitoring in combination with MRI information was evaluated to explore their prognostic potential [107]. After the analysis, they concluded that combining ctDNA and MRI information could be used to predict pCR. As well, Khakoo et al. 2020 found that ctDNA levels during nCRT and post-surgery were associated with the development of metastases [108]. Truelsen et al. (2022) showed that cfDNA levels were higher in LARC patients compared to healthy subjects [109]. In contrast, Ong et al. (2023) found no significant correlation between cfDNA, tumor volume, and tumor regression grade [110].

These results show that cfDNA analysis could be useful in the management of RC, offering a minimally invasive approach for real-time monitoring of treatment response and prognosis, contributing to improving patient outcomes.

RNA analysis can also serve as a biomarker for disease monitoring and response to treatment. There are several subtypes of RNA that could be potentially useful, including messenger RNA (mRNA), microRNA (miRNA), long non-coding RNA (lncRNA), small nuclear RNA (snRNA), circular RNA (circRNA), and piwi-interacting RNA (piRNA) [111]. In terms of stability, mRNA is the least stable, while miRNAs and lncRNAs are more stable, being the most used in RNA analysis in the context of liquid biopsy [111,112].

Recent research has increasingly focused on the role of miRNAs and other noncoding RNAs as potential biomarkers for predicting treatment response in RC. Wada et al. (2021) developed a miRNA panel that effectively predicted response to nCRT, offering a novel tool for assessing treatment outcomes [113]. Similarly, Chen and Wang (2021) explored the impact of serum miRNA levels on nCRT response, identifying hsa-miR-30e as significantly associated with a higher incidence of response among patients compared to non-responders [114]. Their findings emphasize the potential of miRNAs as predictive biomarkers for therapy effectiveness.

Cervena et al. (2021) evaluated miRNA expression in plasma of RC patients before and after therapy [115]. This study showed that miR-122 and miR-142-5p were downregulated in the plasma of RC patients who responded well to treatment. The levels of these miRNAs returned to normal, matching those of healthy individuals, one year after diagnosis. However, in non-responders, these miRNAs remained at low levels.

Li et al. (2020) analyzed levels of LincRNA-p21, a noncoding RNA, in tumor tissue from resected CRC patients and plasma. The major prognostic impact was more significant in RC patients than those with CC. Regarding the plasma levels, high levels of lincRNA-p21 were correlated with shorter OS [116].

These studies demonstrate the potential of RNA analysis in predicting treatment response and prognosis in RC patients.

### 4.3. Extracellular Vesicles

Extracellular vesicles (EVs), first reported by Chargaff and West 1946, are small structures enclosed by a lipidic membrane [117]. EVs are categorized regarding their cellular origin: as exosomes, which originate from the endosomal system, and microvesicles, which are released from the plasma membrane [43,118]. Both healthy and tumor cells secrete EVs, and they are involved in metastasis, angiogenesis, and chemotherapy resistance [118].

EVs serve as valuable biomarkers due to their capacity to transport nucleic acids, lipids, and proteins [43]. However, one of the main challenges in using EV analysis in liquid biopsies is related to the difficulty in isolating these structures since they are susceptible to contamination with non-EV proteins, lipoproteins, and high-density lipoproteins (HDL). There are several methods used in EV isolation that are based on physical and chemical properties and include centrifugation-based isolation, size, affinity, precipitation, or microfluidic techniques [118].

Despite the growing interest in EV research, few studies have focused on the relationship between EVs and prognostic outcomes in RC.

Strybel et al. (2022) conducted a multi-omics study to investigate the proteomic and metabolomic profiles of exosome-derived proteins in RC patients [119]. Their findings suggest that the serum-derived exosome proteome may serve as a potential biomarker for predicting responses to nCRT, as it could distinguish between patients with varying treatment outcomes.

Chen et al. (2024) conducted a study using human plasma small EVs (sEVs) to identify potential biomarkers for nCRT in patients with LARC. Their results demonstrated that differential expression of sEV proteins could distinguish between good and poor responders to treatment. Additionally, they observed significant alterations in the protein composition of sEVs following nCRT, indicating that the treatment induces measurable changes in the EV proteome [120]. This finding complements the study by Strybel et al. (2022) by suggesting not only the predictive value of exosome proteins but also their potential for disease monitoring, allowing clinicians to follow treatment efficacy over time.

Together, these studies emphasize the clinical relevance of exosome proteomics in RC, contributing to personalized treatment strategies based on patient-specific molecular signatures from EVs. These results strengthen the hypothesis that EV-derived proteins may serve as reliable biomarkers for nCRT response, with implications for both pre-treatment prediction and post-treatment monitoring.

## 5. Discussion

Currently, no biomarkers are utilized in clinical practice to predict responses to nCRT. According to ESMO guidelines, rectal cancer staging should include a history and physical examination, digital rectal examination (DRE), full blood count, liver and renal function tests, rigid rectoscopy and preoperative colonoscopy, serum CEA, CT scan of the thorax and abdomen, and pelvic MRI [3].

In a 2017 systematic review and meta-analysis, baseline CEA showed an inverse correlation with DFS, indicating its potential as a prognostic marker in localized and metastatic colorectal cancer. The relationship between baseline CEA and pCR showed significant heterogeneity and publication bias, preventing definitive conclusions. Currently, CEA is routinely employed to assess the response to chemotherapy in patients with metastatic disease, as well as in the ongoing monitoring of high-risk patients during follow-up [121].

MRI is the modality of choice for local staging, treatment planning, and restaging after nCRT [122,123].

Magnetic resonance tumor response grade (mrTRG) is an imaging adaptation from the tumor response grade (TRG) grading system used in histopathology. It is a scale from 1 to 5 to assess the tumor response after nCRT, focusing on the proportions of fibrosis and residual tumor. Recent studies have indicated limitations associated with this grading system, as it may not constitute a consistently reproducible or precise metric. Furthermore, some research has revealed a lack of agreement between pathological TRG and mrTRG. In a recent study regarding RC staging, with the participation of 79 radiologists, the clinical stage matched the histopathological stage in only 34.5%; overstaging was present in 24.5%, and understaging in 16.1% of cases [122,123,124].

Diffusion-weighted imaging (DWI) has enhanced the efficacy of MRI in accurately predicting complete responses. However, MRI remains limited in its ability to detect microscopic tumor cells within fibrotic tissue. Another limitation of MRI is nodal assessment. In cases where a complete response of the primary tumor is achieved, the risk of residual nodal disease may be as high as 17% [123].

Responding tumors also decrease in size after neoadjuvant therapy, and tumor volume reduction rate is an independent prognostic factor. The PROSPECT trial demonstrated that neoadjuvant FOLFOX is non-inferior to nCRT in terms of DFS, OS, rates of local recurrence, complete resection, and pCR. In this clinical trial, patients were restaged by MRI following the completion of neoadjuvant FOLFOX therapy. Those whose primary tumor exhibited a size reduction of less than 20% were subsequently assigned to receive nCRT [123,125].

At present, molecular biomarker testing for RC is restricted to individuals diagnosed with metastatic colorectal cancer (mCRC). Testing for MMR status, *KRAS* and *NRAS* exons 2, 3, and 4, and *BRAF* mutations is recommended in all patients at the time of mCRC diagnosis. These tests will impact the selection of first-line chemotherapy by considering the tumor’s molecular profile in alignment with the principles of precision medicine [126].

Currently, selecting the most effective treatment strategy for LARC is a complex endeavor that poses a significant challenge for the multidisciplinary team involved in patient care.

The identification of a predictive biomarker could facilitate the selection of an optimal therapeutic strategy tailored for each patient.

Several studies on molecular biomarkers were enrolled, some with conflicting results.

This variability in results may stem from structural differences among studies, particularly regarding the nCRT regimen, the time intervals between nCRT and surgery, and the biological tissues being analyzed. To address the heterogeneity seen in studies with small sample sizes, it would be beneficial to initiate a large-scale multicenter study [14]. The clinical relevance of molecular biomarkers in RC is often evaluated in terms of DFS and OS. Variations in neoadjuvant and adjuvant treatments, the quality of the surgical procedure, and, consequently, the characteristics of the surgical specimens also affect these outcomes.

The diverse range of responses to nCRT observed in LARC underscores the intricate relationship between tumor biology and therapeutic response. This variability may be influenced by the interference with multiple molecular pathways, suggesting that identifying a singular predictor may be impractical. A comprehensive predictive model that integrates clinicopathological, radiological, and molecular biomarkers may provide a more robust solution [4,5].

Besides genetic factors, the gut microbiome plays a critical role in maintaining the normal functioning of the digestive system. This highly heterogeneous ecosystem comprises various microorganisms, including bacteria, viruses, fungi, and archaea [127].

Dysbiosis, an imbalance in the gut microbiota, has been closely associated with the development of colon and rectal tumors [128]. The microbiome interacts with the tumor microenvironment, including the immune system, influencing tumor development [129]. Inflammatory processes often arise from disruptions in the gut microbiota, with several bacterial species producing metabolites that induce DNA damage and contribute to inflammation [127,129]. Additionally, the gut microbiome can modulate several molecular pathways involved in cell proliferation, apoptosis, and DNA repair mechanisms, further influencing colorectal carcinogenesis [129].

The microbiome holds promise as a predictive biomarker for therapeutic response. Studies have demonstrated significant alterations in the gut microbiota of patients with locally advanced rectal cancer (LARC), enabling the differentiation between responders and non-responders to neoadjuvant chemoradiotherapy (nCRT) [130,131,132].

The microbiome has also emerged as a promising therapeutic target. Strategies such as dietary interventions, probiotics, and antibiotics are being explored to regulate the gut microbiota, potentially altering the tumor microenvironment, and influencing cancer progression. However, further research is essential to elucidate these complex interactions and develop effective therapeutic approaches [133].

Despite intense research, predictive biomarkers have yet to be identified. To be applied to clinical practice, a biomarker or predictive model must be rigorously validated, easily applicable, and reproducible. These tools could differentiate a priori between good radiotherapy responders and bad responders and spare this last group the disadvantage of delaying surgery [15].

The adaptive phenomena induced by radiation in tumor tissue during nCRT may result in secondary radioresistance. To effectively monitor and profile treatment responses, sequential tumor biopsies or liquid biopsies may offer valuable insights in real time. Liquid biopsies that utilize circulating tumor DNA present a potentially more sensitive method for monitoring treatment responses compared to traditional, anatomical, radiological, and endoscopic evaluations. Additionally, these biopsies could play a crucial role in identifying patients who may be eligible for a W and W strategy, thereby allowing some to forgo surgical intervention [5,15,134].

## 6. Conclusions

This review aimed to explore the rationale underlying potential molecular biomarkers that may predict response to nCRT in LARC, focusing on molecular pathways in RC.

The molecular landscape of RC involves a variety of signaling pathways that contribute to the unique biological behavior of rectal tumors, influencing both prognosis and therapeutic outcomes. Some biomarkers show potential in predicting tumor response to nCRT in RC. Specifically, a relationship has been observed between CIN and a favorable response to nCRT, whereas mutations in *TP53* and *KRAS* are correlated with a less favorable response.

It also must be emphasized the usefulness of ctDNA testing for ctDNA-guided MRD assessment and ctDNA-guided surveillance.

## 7. Future Directions

Further research is required to fully understand the role of these molecular pathways, particularly in RC pathogenesis. These insights may play a significant role in identifying molecular biomarkers that can predict tumor response to neoadjuvant chemoradiotherapy in patients with RC.

The expression of molecular biomarkers in rectal tumors should be studied based on indicators of their response to nCRT. These indicators may include those employed in current clinical practice, such as evaluating MRI radiological responses, assessing tumor regression grade as determined through histopathological examination, and, potentially, in the near future, analyzing ctDNA obtained from liquid biopsies.

In conclusion, more standardized evaluations are necessary to enhance the value of predictive biomarkers in RC.

## Figures and Tables

**Figure 1 biology-13-01007-f001:**
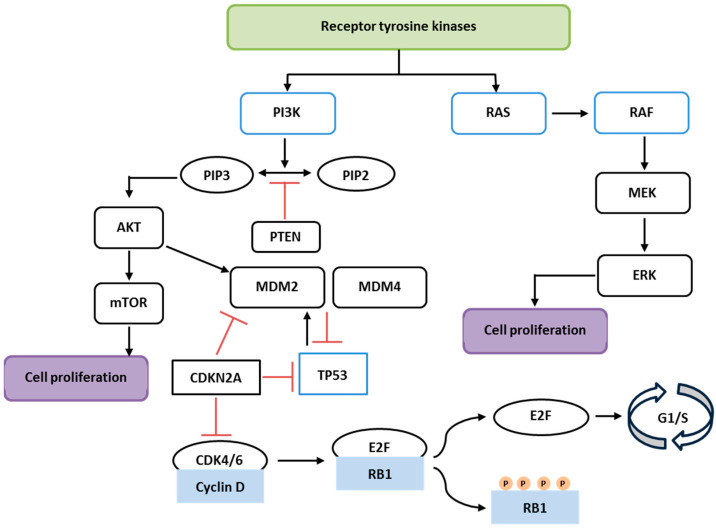
Simplified schematic representation of the PI3K/AKT/mTOR, RAS/RAF/MAPK, RB, and p53 pathways. RTKs activate two major pathways: the PI3K/AKT/mTOR and RAS/RAF/MAPK. PTEN inhibits the PI3K pathway, while TP53 and CDKN2A act as tumor suppressors, blocking cell cycle progression by inhibiting the Cyclin D-CDK4/6 complex and RB1-E2F activation.

**Figure 2 biology-13-01007-f002:**
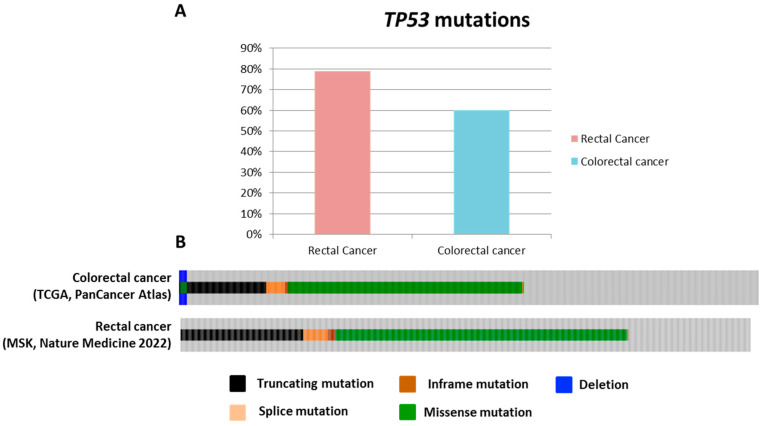
Comparison of *TP53* mutation frequency and types in rectal versus colorectal cancer. (**A**) Frequency of *TP53* mutations in rectal cancer compared to colorectal cancer. (**B**) Types of *TP53* mutations observed in each cancer type. Data source: The Cancer Genome Atlas (TCGA), PanCancer Atlas, and MSK, Nature Medicine 2022—cBioPortal for Cancer Genomics.

**Figure 3 biology-13-01007-f003:**
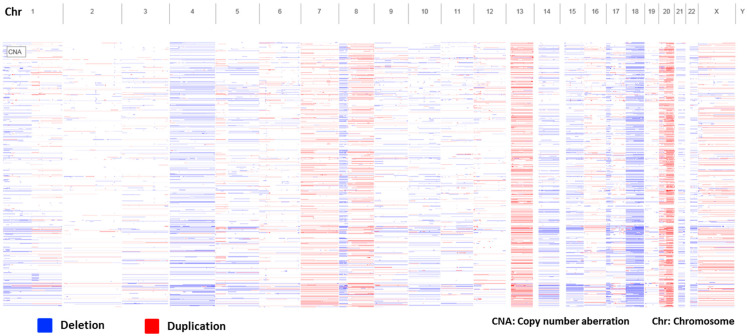
Genomic landscape of rectal cancer showing multiple chromosomal gains (red) and losses (blue) across various chromosomes. Data source: MSK, Nature Medicine, 2022—cBioPortal for Cancer Genomics.

**Figure 4 biology-13-01007-f004:**
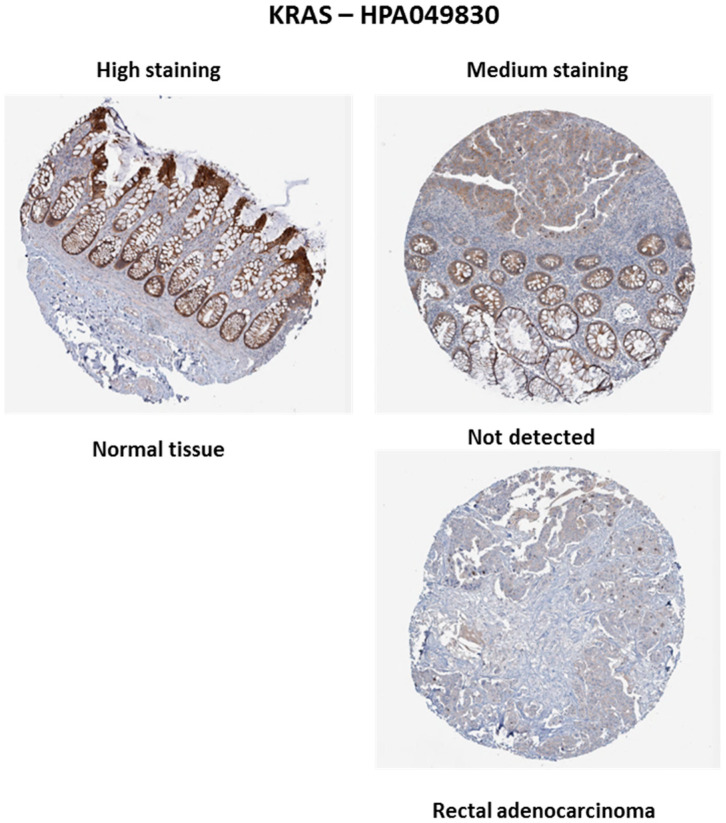
*KRAS* expression in normal and rectal adenocarcinoma tissues. Immunohistochemical staining with the HPA049830 antibody shows high *KRAS* expression in normal tissue, while expression in tumor tissue is medium or undetectable. Scale: 200 µm. Source: The Human Protein Atlas.

**Figure 5 biology-13-01007-f005:**
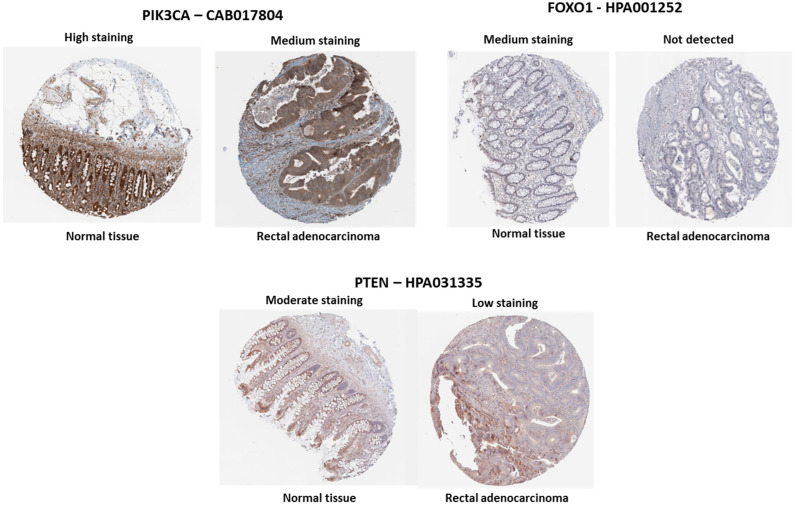
Differential expression of *PIK3CA*, *FOXO1*, and *PTEN* in normal versus rectal adenocarcinoma tissues. Immunohistochemical staining using antibodies CAB017804 (*PIK3CA*), HPA001252 (*FOXO1*), and HPA031335 (*PTEN*). PIK3CA shows high to medium staining in normal tissue and medium staining in rectal adenocarcinoma. *FOXO1* displays medium staining in normal tissue but is undetectable in rectal adenocarcinoma. *PTEN* shows moderate staining in normal tissue and low staining in rectal adenocarcinomas. Scale: 200 µm. Source: The Human Protein Atlas.

**Figure 6 biology-13-01007-f006:**
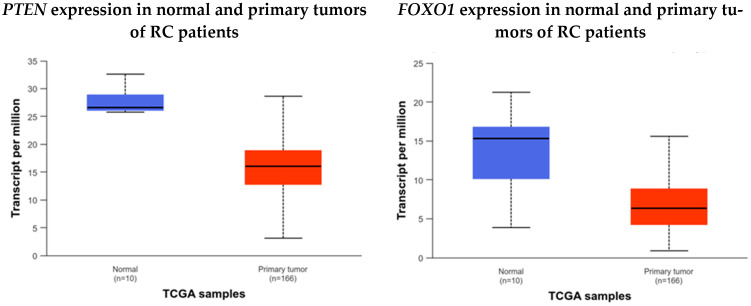
Comparison of *PTEN* and *FOXO1* expression in healthy and rectal cancer patients. The expression of *PTEN* and *FOXO1* is reduced in rectal cancer patients when compared to healthy individuals.

**Figure 7 biology-13-01007-f007:**
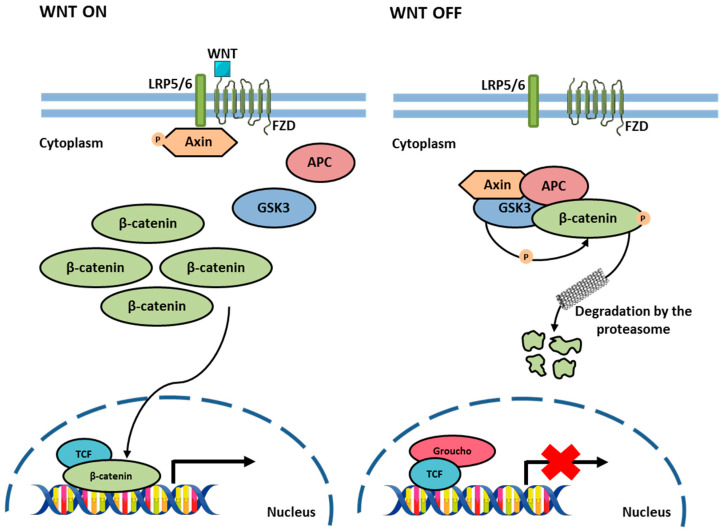
Wnt/β-Catenin Pathway. When the WNT ligands bind to the LRP5/6 and FZD (**left**), the phosphorylation and degradation of β-catenin are inhibited, and then β-catenin accumulates and translocates to the nucleus, where it partners with TCF to activate Wnt-responsive genes. In the absence of WNT ligands (**right**), cytoplasmic β-catenin forms a complex with several proteins, leading to proteasomal degradation. As a result, β-catenin levels remain low in the cytoplasm, inhibiting target gene transcription. Parts of the figure were drawn using Servier Medical Art licensed under a Creative Commons Attribution 3.0 Unported License.

**Figure 8 biology-13-01007-f008:**
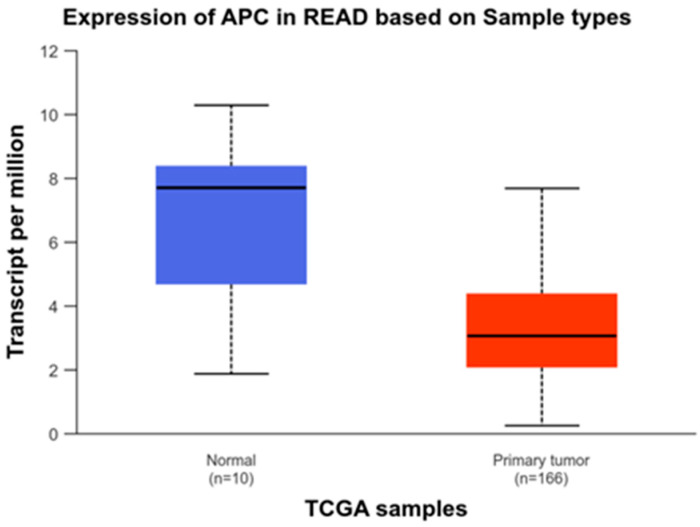
Comparison *APC* expression of healthy and rectal cancer patients, showing a reduced expression in the RC patients’ group.

**Figure 9 biology-13-01007-f009:**
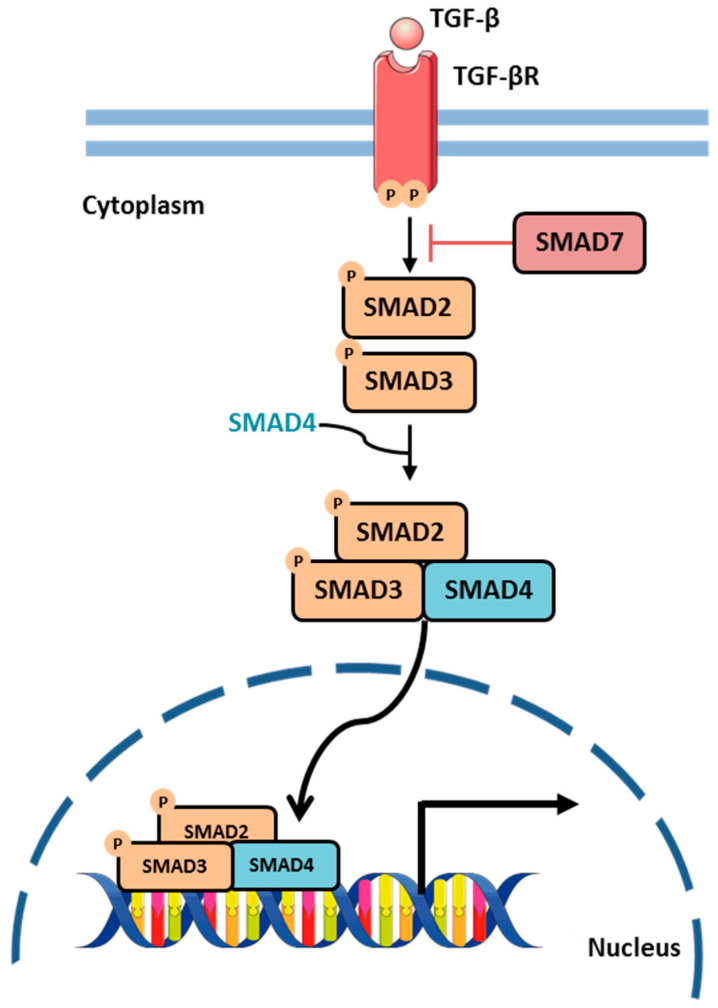
TGF-β signaling pathway in gene expression regulation. Upon TGF-β ligand binding, TGF-β receptor is activated. Then it phosphorylates SMAD2 and SMAD3 (receptor-regulated SMADs). These R-SMADs form a complex with SMAD4 (common mediator SMAD), which translocates to the nucleus to regulate the transcription of target genes. SMAD7 acts as a negative feedback regulator by inhibiting this pathway. Parts of the figure were drawn using Servier Medical Art, licensed under a Creative Commons Attribution 3.0 Unported License.

**Table 1 biology-13-01007-t001:** Rectal cancer biomarkers and their influence on prognosis—a literature review.

Biomarker	Influence in Prognosis	Reference
CIN	Favorable predictor of response to nCRT in LARC.	[61]
	An in vitro study with RC organoids suggests that resistance to radiation is associated with reduced chromosomal instability.	[62]
MSI	MSI-H with loss of MMR and p21WAF1/C1PI expression is predictive of an improved response to neoadjuvant treatment with 5-FU, CPT-11, and radiation therapy.	[63]
	MSI was independently associated with a reduction in pCR for LARC after nCRT.	[24]
	This meta-analysis concluded that there was no significant difference in pCR rate following nCRT in patients with MSI rectal tumors.	[64]
	No prognostic effect of MSI in RC patients.	[21]
	MSI-H was not associated with response to nCRT in patients with RC.	[65]
	Patients diagnosed with LARC and MSI-H showed a high rate of clinical response after immunotherapy.	[66]
CIMP	Three- and five-year DFS was worse in CIMP-positive patients with LARC.	[27]
	CIMP-positive cells may be more sensitive to radiation therapy.	[29]
	This meta-analysis showed no significant association between CIMP and poor outcomes in RC.	[26]
*TP53*	Expression of nuclear p53 protein in rectal carcinoma seems to be a significant predictive factor for local treatment failure after preoperative radiotherapy.	[67]
	*TP53* status is an independent prognostic factor of response to radiotherapy and survival.	[68]
	In patients with an abnormal p53 genotype, overall survival was significantly diminished.	[69]
	The results of this meta-analysis indicate that *TP53* status is a predictive factor response in RC patients undergoing neoadjuvant radiation-based therapy.	[70]
	Patients with *TP53* mutations had worse 5-year progression-free survival (PFS).	[71]
*KRAS*	This meta-analysis showed that the *KRAS* mutation does not affect cancer-specific survival following nCRT and surgery.	[72]
	*KRAS* mutation is associated with lower pCR in LARC.	[73]
	*KRAS* mutational status may affect the prognosis of early RC and may be associated with distant recurrence.	[74]
	Patients with *KRAS* mutations had worse 5-year PFS in LARC.	[71]
	This meta-analysis demonstrated that the KRAS mutation is a predictor of poor prognosis in LARC patients treated with nCRT.	[75]
	The *RAS* mutation seems to be related to poor prognosis and increased risk of recurrence in RC patients undergoing radical surgery after nCRT.	[76]
	*KRAS* mutation is related to decreased DFS and locoregional recurrence-free survival in patients that underwent surgery and following nCRT.	[77]
*BRAF*	Patients with *BRAF*-mutated LARC had shorter PFS and overall survival (OS).	[78]
	*BRAF* mutation predicted poor 5-year OS in patients treated with cetuximab.	[71]
*SMAD4*	Patients with *SMAD4* mutations had shorter PFS.	[78]
*PTEN*	*PTEN* alterations are associated with poor prognosis.	[79]

## Data Availability

Not applicable.

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
