# Peer review of "Rectal Cancer: Exploring Predictive Biomarkers Through Molecular Pathways Involved in Carcinogenesis"

_biology, 2024, doi:10.3390/biology13121007_

Round 1

Reviewer 1 Report

Comments and Suggestions for Authors

This review discusses the potential of molecular biomarkers like CIN, TP53, and KRAS to predict treatment response in locally advanced rectal cancer (LARC) after neoadjuvant chemoradiotherapy (nCRT). While 20% of patients achieve a complete response, 20-38% experience disease progression. Identifying reliable biomarkers could enhance personalized treatment strategies and improve clinical outcomes in the future. Here are some suggestions below:

1. The manuscript provides a comprehensive review of molecular biomarkers in rectal cancer (RC), focusing on their potential to predict response to neoadjuvant chemoradiotherapy (nCRT). The inclusion of both well-established biomarkers and emerging biomarkers is valuable.

2. Clinical examples or case studies demonstrating how these biomarkers are used in practice would be beneficial. This could help contextualize the theoretical discussion and show practical applications for clinicians.

3. Some parts of the manuscript, particularly the sections on where author added the information about CIN and MSI, contain repetitive content. Streamlining these sections would improve the manuscript flow.

4. The conflicting results in the literature, particularly concerning MSI and KRAS mutations, should be discussed in greater depth. Exploring the reasons behind these differences would strengthen the argument for or against their predictive value.

5. The overall manuscript is well-designed and provide a valuable information in the field of healthcare.

Author Response

Reviewer 1

This review discusses the potential of molecular biomarkers like CIN, TP53, and KRAS to predict treatment response in locally advanced rectal cancer (LARC) after neoadjuvant chemoradiotherapy (nCRT). While 20% of patients achieve a complete response, 20-38% experience disease progression. Identifying reliable biomarkers could enhance personalized treatment strategies and improve clinical outcomes in the future. Here are some suggestions below:

  1. The manuscript provides a comprehensive review of molecular biomarkers in rectal cancer (RC), focusing on their potential to predict response to neoadjuvant chemoradiotherapy (nCRT). The inclusion of both well-established biomarkers and emerging biomarkers is valuable.

Author's Response: Thank you for your valuable comments. We have meticulously addressed each concern raised by the reviewer.

  1. Clinical examples or case studies demonstrating how these biomarkers are used in practice would be beneficial. This could help contextualize the theoretical discussion and show practical applications for clinicians.

Author's Response: Thank you for your thoughtful recommendations. We have taken your suggestions into account and have integrated the following sentences into the Discussion section: “Currently, no biomarkers are utilized in clinical practice to predict responses to nCRT. According to ESMO guidelines, rectal cancer staging should include a history and physical examination, digital rectal examination (DRE), full blood count, liver and renal function tests, rigid rectoscopy and preoperative colonoscopy, serum CEA, CT scan of the thorax and abdomen, and pelvic MRI [3].  In a 2017 systematic review and meta-analysis, baseline CEA showed an inverse correlation with DFS, indicating its potential as a prognostic marker in localized and metastatic colorectal cancer. The relationship between baseline CEA and pCR showed significant heterogeneity and publication bias, preventing definitive conclusions. Currently, CEA is routinely employed to assess the response to chemotherapy in patients with metastatic disease, as well as in the ongoing monitoring of high-risk patients during follow-up [120].  MRI is the modality of choice for local staging, treatment planning, and restaging after nCRT [121,122]. Magnetic resonance tumor response grade (mrTRG) is an imaging adaptation from the tumor response grade (TRG) grading system used in histopathology. It is a scale from 1 to 5 to assess the tumor response after nCRT, focusing on the proportions of fibrosis and residual tumor. Recent studies have indicated limitations associated with this grading system, as it may not constitute a consistently reproducible or precise metric. Further-more, some research has revealed a lack of agreement between pathological TRG and mrTRG. In a recent study regarding RC staging, with the participation of 79 radiologists, the clinical stage matched the histopathological stage in only 34.5%; overstaging was present in 24.5%, and understaging in 16.1% of cases [121–123]. Diffusion-weighted imaging (DWI) has enhanced the efficacy of MRI in accurately predicting complete responses. However, MRI remains limited in its ability to detect microscopic tumor cells within fibrotic tissue. Another limitation of MRI is nodal assessment. In cases where a complete response of the primary tumor is achieved, the risk of residual nodal disease may be as high as 17% [122]. Responding tumors also decrease in size after neoadjuvant therapy and tumor volume reduction rate is an independent prognostic factor. The PROSPECT trial demonstrated that neoadjuvant FOLFOX is non-inferior to nCRT in terms of DFS, OS, rates of local recurrence, complete resection, and pCR. In this clinical trial, patients were restaged by MRI following the completion of neoadjuvant FOLFOX therapy. Those whose primary tumor exhibited a size reduction of less than 20% were subsequently assigned to receive nCRT [122,124].  At present, molecular biomarker testing for RC is restricted to individuals diagnosed with metastatic colorectal cancer (mCRC). Testing for MMR status, KRAS and NRAS exons 2, 3, and 4, and BRAF mutations is recommended in all patients at the time of mCRC diagnosis. These tests will impact the selection of first-line chemotherapy by considering the tumor's molecular profile in alignment with the principles of precision medicine [125].”

  1. Some parts of the manuscript, particularly the sections on where author added the information about CIN and MSI, contain repetitive content. Streamlining these sections would improve the manuscript flow.

Author's Response: We have reviewed this section as requested.

  1. The conflicting results in the literature, particularly concerning MSI and KRAS mutations, should be discussed in greater depth. Exploring the reasons behind these differences would strengthen the argument for or against their predictive value.

Author's Response: We thank the reviewer for their suggestion. We have attempted to answer this question in the Discussion section: “Several studies on molecular biomarkers were enrolled, some with conflicting results. This variability in results may stem from structural differences among studies, particularly regarding the nCRT regimen, the time intervals between nCRT and surgery, and the biological tissues being analyzed. To address the heterogeneity seen in studies with small sample sizes, it would be beneficial to initiate a large-scale multicentre study [14]. The clinical relevance of molecular biomarkers in RC is often evaluated in terms of DFS and OS. Variations in neoadjuvant and adjuvant treatments, the quality of the surgical procedure, and, consequently, the characteristics of the surgical specimens also affect these outcomes.”

  1. The overall manuscript is well-designed and provide a valuable information in the field of healthcare.

Author's Response: Thank you for your insightful contributions.

Reviewer 2 Report

Comments and Suggestions for Authors

Locally advanced rectal cancers are treated with chemoradiotherapy before surgery to shrink tumors, but around 20-38% of patients experience residual disease or tumor growth, highlighting the need for methods to predict response. Sheila Martins et al. explore potential molecular biomarkers to identify responders and non-responders to chemoradiotherapy, with liquid biopsies possibly playing a key role in future evaluations.

The microbiome, including the bacterial composition within the tumor microenvironment, plays a significant role in the development, progression, and response to therapy in rectal cancer. Recent research has revealed that the gut microbiota—the community of bacteria, fungi, and other microorganisms residing in the gastrointestinal tract—can influence tumor biology and treatment outcomes. The interplay between the microbiome and rectal cancer pathways is complex, involving immune modulation, inflammation, genetic alterations, and metabolic changes that promote tumor growth and therapy resistance. This multifaceted relationship affects various biological pathways, influencing tumor progression within the microenvironment. Therefore, it is essential to discuss the tumor’s progression under the influence of the microbiome and its impact on therapeutic strategies.

Genetic mutations are primary drivers of cancer development, progression, and response to therapy. In rectal cancer, several mutations are commonly observed, and understanding their implications is essential for predicting clinical outcomes and shaping therapeutic strategies. However, it is equally important to consider the pathways and gene alterations at the transcriptomic and epigenetic levels. The combined analysis of the genetic, transcriptomic, and epigenetic landscapes of rectal cancer provides valuable insights into the molecular mechanisms underlying tumorigenesis, progression, and treatment response. A comprehensive understanding of these molecular layers is critical for developing more effective and personalized therapeutic approaches.

In the Liquid Biopsy section, the authors summarized three key components. However, it is essential to provide more detailed information about the underlying technologies involved. Expanding on the various techniques and their applications will offer a clearer understanding of how liquid biopsy is used for early detection, monitoring treatment response, and tracking tumor progression in rectal cancer and other cancers.

Author Response

Reviewer 2

Locally advanced rectal cancers are treated with chemoradiotherapy before surgery to shrink tumors, but around 20-38% of patients experience residual disease or tumor growth, highlighting the need for methods to predict response. Sheila Martins et al. explore potential molecular biomarkers to identify responders and non-responders to chemoradiotherapy, with liquid biopsies possibly playing a key role in future evaluations.

Authors' Response: Thank you very much for your comments. We have addressed each of the reviewers' concerns individually.

  1. The microbiome, including the bacterial composition within the tumor microenvironment, plays a significant role in the development, progression, and response to therapy in rectal cancer. Recent research has revealed that the gut microbiota—the community of bacteria, fungi, and other microorganisms residing in the gastrointestinal tract—can influence tumor biology and treatment outcomes. The interplay between the microbiome and rectal cancer pathways is complex, involving immune modulation, inflammation, genetic alterations, and metabolic changes that promote tumor growth and therapy resistance. This multifaceted relationship affects various biological pathways, influencing tumor progression within the microenvironment. Therefore, it is essential to discuss the tumor’s progression under the influence of the microbiome and its impact on therapeutic strategies.

Authors' Response: We appreciate the recommendation and have incorporated the following sentences into the introduction section: "Besides genetic factors, the gut microbiome plays a critical role in maintaining the normal functioning of the digestive system. This highly heterogeneous ecosystem comprises various microorganisms, including bacteria, viruses, fungi, and archaea [126]. Dysbiosis, an imbalance in the gut microbiota, has been closely associated with the development of colon and rectal tumors [127]. The microbiome interacts with the tumor microenvironment, including the immune system, influencing tumor development [128]. Inflammatory processes often arise from disruptions in the gut microbiota, with several bacterial species producing metabolites that induce DNA damage and contribute to inflammation [126,128]. Additionally, the gut microbiome can modulate several molecular pathways involved in cell proliferation, apoptosis, and DNA repair mechanisms, further influencing colorectal carcinogenesis [128]. The microbiome holds promise as a predictive biomarker for therapeutic response. Studies have demonstrated significant alterations in the gut microbiota of patients with locally advanced rectal cancer (LARC), enabling the differentiation between responders and non-responders to neoadjuvant chemoradiotherapy (nCRT) [129–131]. The microbiome has also emerged as a promising therapeutic target. Strategies such as dietary interventions, probiotics, and antibiotics are being explored to regulate the gut microbiota, potentially altering the tumor microenvironment and influencing cancer progression. However, further research is essential to elucidate these complex interactions and develop effective therapeutic approaches [132]”.

  1. Genetic mutations are primary drivers of cancer development, progression, and response to therapy. In rectal cancer, several mutations are commonly observed, and understanding their implications is essential for predicting clinical outcomes and shaping therapeutic strategies. However, it is equally important to consider the pathways and gene alterations at the transcriptomic and epigenetic levels. The combined analysis of the genetic, transcriptomic, and epigenetic landscapes of rectal cancer provides valuable insights into the molecular mechanisms underlying tumorigenesis, progression, and treatment response. A comprehensive understanding of these molecular layers is critical for developing more effective and personalized therapeutic approaches.

 Authors' Response: We thank the reviewer for their insightful remark and have added the following sentences to Section 3: Predictive and Prognostic Biomarkers: “The integration of this genetic data with transcriptomic and epigenetic analyses in rectal cancer research provides a comprehensive view of the molecular mechanisms underlying tumor development, progression, and treatment response, ultimately leading to the characterization of robust biomarkers [79,80]. In a recent pilot study, Ci-calini et al. 2024 demonstrated the potential of multi-omics analysis to predict CRT re-sponse in patients with LARC [80]. Boldrini et al. (2024) combines gut microbiota and ctDNA information to explore the predictive performance. Although this study is still in its early stages and no re-sults have been published to date, it presents a promising and groundbreaking ap-proach. By integrating the various molecular levels and microbiome data as potential biomarkers, this study results can improve patient’s outcomes [81]. Jiang et al. (2022) further demonstrated the potential of multi-omics approaches to identify predictive biomarkers, utilizing genomic DNA and cell-free DNA (cfDNA). Their findings revealed that increased expression of genes such as EGFR and HSP90AA1 was associated with higher sensitivity to chemotherapy, while copy num-ber losses were predominantly observed in patients with poorer prognoses. Im-portantly, EGFR and HSP90AA1 exhibited significant differences across multiple mo-lecular layers, demonstrating their potential as key biomarkers for prognosis and treatment response [82]. The development of multi-omics approaches offers a unique opportunity to un-ravel the complex landscape of rectal cancer, clarifying the complex interactions of ge-netic, transcriptomic, metabolomic and epigenetic factors. By integrating diverse layers of molecular data, these approaches not only enhance our understanding of predictive biomarkers but also pave the way for the development of more precise diagnostic tools and personalized therapeutic strategies”.

  1. In the Liquid Biopsy section, the authors summarized three key components. However, it is essential to provide more detailed information about the underlying technologies involved. Expanding on the various techniques and their applications will offer a clearer understanding of how liquid biopsy is used for early detection, monitoring treatment response, and tracking tumor progression in rectal cancer and other cancers.

Authors' Response: We thank the reviewer for their suggestion and have added the following sentences to Section 4: Liquid Biopsy: “NGS allows the detection of a wide range of genomic alterations. However, its use in clinical practice is challenging due to the long turnaround time and associated costs. When compared to NGS, PCR-based technologies only enable the targeted analysis, with a low cost and short turnaround time [86].”, “Regarding the use of CTCs in clinical practice, the Food and Drug Administration has approved The CellSearch™ by Veridex. This method has been used in the clinical practice in patients with metastatic breast, prostate and colorectal cancer [96]”, “Since ctDNA constitutes only a small fraction of cfDNA, highly sensitive techniques are required to identify tumor-specific alterations. Mutations in ctDNA can be detected by PCR or NGS. However, PCR-based methodologies only detect mutations in specific genes, limiting their utility. On the other hand, NGS approaches offer high coverage and can be applied in a genome-wide manner [103,104]. In addition to these methods, methylation assays are highly applicable to detect methylation patterns in specific regions. The use of MRE-seq appears to be a promising methodology. Kwon et al. (2023) demonstrated the applicability of this approach, showing high accuracy in diagnosing and detecting global hypomethylation patterns in lung and colorectal Cancer. In rectal cancer, Shalaby et al. (2017) demonstrated the ability of MGMT and ERCC1 methylation status to distinguish between benign and malignant rectal tumors [105]”.

Round 2

Reviewer 2 Report

Comments and Suggestions for Authors

The authors addressed all my questions and revised the paper accordingly.